# Human Antimicrobial Use in Bangladesh: Five-Year Trend Analysis Including COVID-19 Pandemic Era

**DOI:** 10.3390/antibiotics14090868

**Published:** 2025-08-28

**Authors:** S. M. Sabrina Yesmin, Paritosh Chakma, Umme Habiba, Anders Rhod Larsen, Terence Tino Fusire, Sangay Wangmo, Shila Sarkar, Majda Attauabi

**Affiliations:** 1Directorate General of Drug Administration, Aushad Bhavan, Mohakhali, Dhaka 1212, Bangladesh; 2World Health Organization Country Office for Bangladesh, Dhaka 1212, Bangladesh; chakmap@who.int (P.C.); fusiret@who.int (T.T.F.); wangmos@who.int (S.W.); sarkarsh@who.int (S.S.); 3Statens Serum Institut, 2300 Copenhagen, Denmark; arl@ssi.dk (A.R.L.); maat@ssi.dk (M.A.)

**Keywords:** antimicrobial use, human antimicrobial, COVID-19, pandemic, Bangladesh

## Abstract

**Background:** This paper provides the first national analysis of antimicrobial use (AMU) of oral and parenteral dosages in Bangladesh, as well as biannual trends for the years from 2019 to 2023. It also analyzes the effect of the COVID-19 pandemic on AMU. **Methods:** AMU was analyzed in accordance with the WHO Anatomical Therapeutic Chemical classification and defined daily doses per 1000 inhabitants per day methodology. Data on antimicrobial medicine dispatched from manufacturers’ central warehouse was collected and categorized based on the WHO’s Access, Watch, and Reserve (AWaRe) classification. **Findings:** This AMU surveillance demonstrates an increase in the use of antimicrobial medicines from 2021 to 2022, and in 2023, it decreased, with our national AMU surveillance data indicating that cefixime and azithromycin were the most consumed antibiotics during this period. Most antibiotics used in Bangladesh are broad-spectrum ‘Watch’-category antibiotics. Among oral antibiotics, 50 to 67% are from the ‘Watch’-category. When considering only parenteral antibiotics, 70 to 91 % fall under the ‘Watch’-category. Third-generation cephalosporin consumption has been found to be higher than second- and first-generation cephalosporins. The oral antimicrobials are more commonly used than parenteral ones. AMU notably increased during the COVID-19 pandemic, especially in the case of systemic antibacterial use. **Conclusions:** To achieve the global target of 70% use of Access category antibiotics by 2030, the use of Watch-group antibiotics, like cefixime, azithromycin, ciprofloxacin, levofloxacin, and ceftibuten, needs to be reduced through investing in and strengthening stewardship programs and eliminating self-medication in Bangladesh. The findings of this study provide useful information to policymakers to tackle AMR in Bangladesh.

## 1. Introduction

Antimicrobial resistance (AMR) is considered a silent pandemic, with estimates of yearly deaths reaching 1.27 million [1]. AMR has resulted in increasingly ineffective treatment of infections in different regions of the world [2]. The emergence of resistance in any country or part of the world can quickly develop into a worldwide problem [3]. Antimicrobial use (AMU) is one of the main factors driving AMR, and epidemiological studies have established a direct link between antibiotic overuse or improper use and the spread or emergence of resistant bacterial strains globally [4]. Reducing AMR in low- and middle-income countries (LMICs) requires preventing misuse, ensuring access to essential antimicrobials, and systemic monitoring of AMU to identify areas for improvement [5].

The Global Action Plan of the World Health Organization (WHO) suggests establishing a framework that would monitor and report on the use of antimicrobials and their effect on human health [6]. The 2001 AMR strategy outlined by the WHO contemplates monitoring of resistance, AMU, and disease burden [7]. The National Strategy and Action Plan for Antimicrobial Resistance Containment in Bangladesh (2023–2028) also focuses on the implementation of AMU surveillance in Bangladesh [8]. This paper represents Bangladesh’s first AMU surveillance report for antimicrobials for human use. Veterinary antimicrobials are excluded. 

As the national medicine regulatory authority in Bangladesh, the Directorate General of Drug Administration (DGDA) was designated as the national center for monitoring AMU surveillance in Bangladesh. The DGDA—with technical support from WHO Bangladesh [9]; the Fleming Fund Fellowship Program [10]; Statens Serum Institut, Denmark [11]; and the International Livestock Research Institute, Kenya [12]—developed an AMU surveillance system based on the WHO’s guidelines [13] for monitoring AMU in Bangladesh and joined WHO’s Global Antimicrobial Resistance and Use Surveillance System (GLASS) platform in 2022.

In this study, data were analyzed by the WHO’s Access, Watch, and Reserve (AWaRe) classification. The AWaRe classification can improve antibiotic stewardship and optimal use. Stewardship programs can closely monitor the Watch antibiotic group, which potentially drives more resistance than the antibiotics of the Access group [14].

The outbreak of coronavirus disease 2019 (COVID-19) was declared as a pandemic by the WHO from 11 March 2020 to 5 May 2023 [15]. This study covers the periods before (2019), during (2020 to June 2023), and after (July to December 2023) the COVID-19 pandemic. The COVID-19 pandemic exacerbated the pre-existing burden of AMR, with antibiotics being utilized beyond their intended use [16]. A decrease in antimicrobial use during the COVID-19 pandemic in 2020 was noticed in high-income countries like those in the European Union, England, and Canada [17]. Saleem Z et al. [18] suggested that the lack of information on antibiotic use in LMICs revealed gaps that need to be urgently filled. Mondal UK et al. [19] confirmed excessive use of antibiotics among Bangladeshi physicians treating COVID-19 patients, irrespective of disease severity. This study aimed to assess the trends of AMU and COVID-19-associated changes, specifically examining antimicrobial usage volume and pattern, in Bangladesh. 

## 2. Results

### 2.1. National Antimicrobial Use Pattern of Bangladesh (2019 to 2023) 

Table 1 shows the overall national AMU (27.55 to 62.36 defined daily doses (DDDs) per 1000 inhabitants per day (DID)) from 2019 to 2023. Bangladesh’s national AMU data shows a high increasing trend from 2021 to 2022, and in 2023, it decreased. The oral route of administration is the most common.

### 2.2. Antimicrobial Use at Second Level of Anatomical Therapeutic Chemical Classification 

Figure 1 shows data on AMU in Bangladesh from 2019 to 2023 at the second level of the Anatomical Therapeutic Chemical (ATC) classification. An increase in the use of antibacterials for systemic use (ATC-J01) starting from July to December 2020 was noticed and represented 90% of national AMU during the COVID-19 outbreak. Coinciding with the COVID-19 pandemic, a higher Antiprotozoal (ATC-P01) use rate was detected (7.26 DID in January to June 2022 compared to 1.97 DID in July to December, 2019). The use of antivirals for systemic use (ATC-J05) was also observed, with a modest use of up to 0.22 DID (July to December 2021). It is evident that antivirals for systemic use were not widely used to treat patients with COVID-19 in Bangladesh.

### 2.3. Antimicrobial Use at Third Level of Anatomical Therapeutic Chemical Classification 

Figure 2 shows data on antimicrobial use at the third level of the ATC classification. A high consumption of other beta-lactam antibacterials (J01D) was observed during the COVID-19 pandemic period starting from July to December 2020. Also, macrolides, lacosamide, and streptogramins (J01F); quinolone antibacterials (J01M); and agents against amoebiasis and other protozoal diseases (P01A) showed an increasing trend in this period.

### 2.4. Drug Utilization 75% for Oral Dosage Form of Antimicrobials 

Table 2 shows Drug Utilization 75% (DU 75) for oral route of administration, which corresponds to the most consumed antimicrobials covering 75% of national antimicrobial use. The results showed that ten molecules constituted 75% of consumption across the five years studied (2019 to 2023); among them, six generics (cefixime, azithromycin, cefuroxime and beta-lactamase inhibitor, ciprofloxacin, metronidazole, and doxycycline) consistently contributed to each period.

### 2.5. Drug Utilization 75% for Parenteral Dosage Form of Antimicrobials 

Table 3 shows DU 75 for the parenteral route of administration. Ceftriaxone was the most consumed antimicrobial for the parenteral route of administration, with increasing consumption during the COVID-19 pandemic. Moxifloxacin use also showed a markedly increasing trend during the COVID-19 pandemic period; however, before January to June 2021, it was not among the DU75-medicins. 

### 2.6. Use of Cephalosporin Group Antibiotics

Figure 3 shows the trend regarding cephalosporin use in Bangladesh from 2019 to 2023. Among the cephalosporin groups, third-generation cephalosporins exhibit the highest level of consumption both for the oral and parenteral routes of administration.

### 2.7. Antibiotic Use Trend According to WHO-AWaRe Classification of Antibiotics 

Figure 4 shows the trend of oral antibiotic use according to the WHO AWaRe classification in Bangladesh from 2019 to 2023. It is evident that the Watch-category antibiotics reached their peak of consumption in Bangladesh, estimated at 50 to 67%. Access antibiotic consumption showed a peak of 22 to 37%, and Reserve antibiotics had a peak of 0.14 to 0.24% for the oral route of administration. In the COVID-19 pandemic, the Watch-category oral antibiotics showed an increase in consumption of 169.64% (January to June 2021) compared to the pre-COVID-19 pandemic period (July to December 2019). This surge is attributed to an increase in the use of cefixime (J01DD08), azithromycin (J01FA10), and ciprofloxacin (J01MA02), as shown in Table 2.

Figure 5 shows the trend of parenteral antibiotic use according to the WHO AWaRe classification in Bangladesh from 2019 to 2023. Watch-category antibiotics were the most consumed antibiotics (70 to 91%) for the parenteral route of administration. Access-category antibiotics constituted 9 to 30%, and the Reserve-category range was from 0.22 to 0.63% for the parenteral route of administration. 

## 3. Discussion

AMR in Bangladesh is more of a social issue than a medical or health issue due to the easy availability of antimicrobials over the counter (OTC) and their irrational use [20]. The WHO Report on antibiotic consumption surveillance (2016–2018) states that monitoring AMU is essential for national and local stewardship to pinpoint areas for improvement and facilitate targeted interventions against AMR [21]. However, no country-wide AMU surveillance mechanism existed in Bangladesh. This is the first antimicrobial use surveillance report covering 2019 to 2023 by measuring DID according to the ATC classification system. This study shows that the overall national AMU ranged from 27.55 to 62.36, according to defined daily dose per 1000 inhabitants per day, from 2019 to 2023.

The results of our study highlight key findings regarding AMU patterns during the COVID-19 pandemic in Bangladesh. A high increasing trend of AMU was observed from 2021 to 2022 during the COVID-19 pandemic period, and in 2023, it decreased. Although the trend after the COVID-19 pandemic (July to December 2023) showed a decreasing pattern, comparison with a longer post-COVID-19 period may provide a more comprehensive outline of change patterns.

According to this study, consumption of antibacterials for systemic use notably increased during the COVID-19 pandemic, and it constituted 90% of national antimicrobial use from January to June 2021 in Bangladesh. The use of other beta-lactam antibacterials (J01D) was high during the COVID-19 pandemic period. A meta-analysis by Khan S et al. [22] on the use of antimicrobials by hospitalized patients with COVID-19 reported that 69% of patients received at least one course of antibiotics. Antibiotic use during the COVID-19 pandemic was influenced by multiple factors; for example, many patients with COVID-19 were treated empirically with antibiotics due to fear of bacterial co-infections, and evidence shows that the actual bacterial co-infection rate was much lower than the rate of antibiotic use [23]. A study by Malik SS et al. [24] identified that lack of diagnostic facilities and limited access to confirmatory laboratory tests in Bangladesh further encouraged empirical antibiotic prescribing, leading to the overuse of antibiotics. AMR increased when antimicrobials were used empirically among patients with COVID-19. Underlying health conditions such as diabetes, hypertension, and other chronic diseases are prevalent in the population of Bangladesh, and this factor may have contributed to increased antibiotic use as a preventive measure during COVID-19 infections [25]. A study by Sumon SA et al. [26] shows that physicians’ fundamental knowledge of Antimicrobial Stewardship Programs (ASPs) and rational antibiotic prescription was found to fall short of the standard in Bangladesh. Physicians aware of ASPs were 33% less likely to wait for laboratory results before prescribing antibiotics.

It is also important to consider the antiviral use pattern to provide a broader understanding about antimicrobial management during pandemic. In this study, it is shown that antivirals for systemic use were not widely used to treat patients with COVID-19 in Bangladesh. This pattern shows potential gaps in the management of viral infection and indicates that a systematic approach is mandatory for implementing antimicrobial stewardship and infectious disease management [27]. Although national guidelines on clinical management of COVID-19 were available [28], consumption of antivirals seemed low, indicating a need for further exploration of reasons for the lower usages and strategies to promote their appropriate usage. 

The misuse of antibiotics might be even more obvious in LMICs and resource-limited areas which are less prepared for pandemics. Healthcare providers in these areas have few diagnostic and treatment options, and consequently, have no other choice than to utilize antibiotics in the treatment of COVID-19 [29]. In the COVID-19 pandemic, increased infection prevention and control (IPC), decreased international travel, and decreased elective hospital procedures may have reduced AMR pathogen selection and spread in the short term, but the opposite effects may be seen if antibiotics are more widely used, misused, or overused [30]. 

This raises concerns about the rationality of prescribing practices during public health emergencies. The observed rise in the use of certain antimicrobials highlighted in this study, including those with an oral route of administration (such as cefixime, cefuroxime and beta-lactamase inhibitor, ciprofloxacin, metronidazole, and doxycycline) and those with a parenteral route of administration (such as ceftriaxone and moxifloxacin), during the pandemic raises questions with respect to the rationality of antibiotic prescribing practices. Among the cephalosporine group, third-generation cephalosporine showed the highest level of consumption for both the oral and parenteral routes of administration. A study by Parveen M et al. [31] showed that the practice of irrational antibiotic prescribing and self-medication was relatively high among COVID-19-positive participants in Bangladesh.

From this study, it can be observed that Watch-category antibiotics reached their peak of consumption in Bangladesh between 2019 and 2023, ranging from 50 to 67% for the oral antibiotics and when considered only parenteral antibiotics 71 to 91% fall under the Watch-category. Watch-category antibiotic consumption persisted throughout during the COVID-19 pandemic. This spike was mainly caused by the overuse of specific antibiotics like cefixime, azithromycin, and ciprofloxacin. These findings indicate that monitoring and regulating the use of the ‘Watch’-category antibiotics is urgently needed to curb AMR in Bangladesh [32]. The recent (9 September 2024) UN General Assembly goal is to ensure the global target of at least 70% use of Access-group antibiotics by 2030 [33]. However, our findings show that Bangladesh still relies heavily on Watch-group antibiotics. This study highlighted that Access-category antibiotics constitute a range of 22 to 37% for the oral route of administration and 9 to 30% for the parenteral route of administration in Bangladesh. The UN General Assembly goal can be achieved by reducing the use of the most consumed Watch-group antibiotics, like cefixime, azithromycin, ciprofloxacin, levofloxacin, and ceftibuten, through investing in and strengthening stewardship programs. High consumption of Watch-category antibiotics, particularly in LMICs, leads to challenges in antibiotic stewardship [34].

In this context, strengthening AMU surveillance becomes a key strategic tool. AMU surveillance can facilitate the decision-making processes of health policymakers and prescribers and, thus, help monitor the impact of national actions to optimize the access and rational use of antimicrobials [35]. Although this study uses national-level central distribution data, future AMU surveillance initiatives should aim for categorical disaggregation by demographics, region, healthcare sector, and patient setting to better inform targeted stewardship interventions.

## 4. Materials and Methods

### 4.1. Data Sources

In Bangladesh, 98% of medicines are locally manufactured. There are 321 medicine manufacturing companies in Bangladesh, of which 253 are functional, 59 are suspended, 3 are non-functional, and 6 have had to pause production. Among the functional companies, 94 manufactured 132 generic antimicrobial medicines, among which 96 are antibiotics [36]. Table 4 shows the number of antimicrobial generic medicines for which data were collected for this surveillance study on AMU in Bangladesh. 

Marketing authorization holders and manufacturers are the same in Bangladesh. Thus, distribution data were collected from such companies, excluding export volume. We used a data triangulation approach to ensure the distributed antimicrobials were neither expired nor taken or returned. We instructed the manufacturers/distributors to submit their antimicrobial distribution data, excluding data on returned, recalled, and destroyed products. Additionally, data were collected after each calendar year to incorporate final reconciliation with destruction and recall records. We also cross-checked with IQVIA sales data.

### 4.2. Methods

We used the WHO GLASS methodology for national surveillance of AMC. According to the updated 2022 GLASS report, the classical distinction between AMC and AMU has been revised to provide a more unified terminology. As per the updated framework, “AMU” encompasses both volume-based (previously AMC) and clinical-level (previously AMU) data. Our data fall under “medical-level AMU” (m-AMU), which indicates volume estimates of antimicrobials distributed but not linked to patient-level clinical information, and these data sources provide a proxy estimate of use of antimicrobials [37,38].

Aggregated national-level biannual public and private sector distribution data for oral solid, liquid, powder for suspension, and parenteral dosage forms were collected. Oral solid and parenteral dosage forms were collected as pieces (pcs) based on the strength, whereas liquid dosage forms were collected based on pack size unit volume. Data were collected according to the ATC level 5 and included active substance(s), label, pack size, pack size unit, administration route, strength, strength per unit, salt content, and other chemical substance combinations. 

Antimicrobials were classified according to the ATC classification system, and national AMU for oral and parenteral dosage was calculated according to the defined daily doses (DDDs) per 1000 inhabitants per day (DID) methodology. Data on antimicrobial medicine dispatched from manufacturers’ central warehouses was collected and categorized based on WHO’s AWaRe classification.

### 4.3. Population Data

Bangladesh’s population data were obtained from “World Population Prospects 2024”, published by the United Nations Department of Economic and Social Affairs, as per the WHO GLASS methodology. Denominators taken into consideration for January to June were ‘Total Population as of 1 January (thousands)’ and, for July to December, ‘Total Population as of 1 July (thousands).’ Real calendar days were utilized for both periods in the specified year.

### 4.4. Calculation of Defined Daily Dose (DDD) and Defined Daily Dose per 1000 Inhabitants per Day (DID) [13]

“WHO GLASS AMC Excel template 2024, version 4” was used to validate individual package data and to obtain individual defined package portion (DPP) values (number of DDDs contained in one medicinal product package (MPP)).$$DPP=\frac{PACKCONTENT\;}{DDD}$$

After obtaining DPP, DDD packages were calculated by multiplying with six monthly total packages, and DID was calculated using the following formula: $$DDD=\frac{Total\;grams\;used}{WHO\;DDD\;value\;in\;grams}$$$$DID=\frac{Total\;DDD\;}{Population\times actual\;days\;of\;six\;months\;period\;of\;that\;year}\times 1000$$

### 4.5. Analyzing Effect of COVID-19 Pandemic on National AMU 

This study compared data pertaining to before (from January to December 2019), during (from January 2020 to June 2023) and after (July 2023 to December 2023) the COVID-19 pandemic period to analyze the effect of the COVID-19 pandemic on AMU. 

## 5. Conclusions

This study provides the first nationwide, five-year trend analysis of human AMU in Bangladesh, covering data from before, during, and immediately after the COVID-19 pandemic. The findings derived from this study reveal a concerning pattern of increased AMU, especially during the pandemic years, driven largely by the overuse of ‘Watch’-category antibiotics such as cefixime, azithromycin, and ciprofloxacin. The disproportionate reliance on Watch-group antibiotics, compared to the global targeting of Access-group antibiotic use, highlights significant gaps in antimicrobial stewardship and prescribing practices.

Given the demonstrated trends, strengthening national AMU surveillance systems is critical to support evidence-based policy decisions and to track progress towards national and global AMR targets. Future research should also look at longer post-pandemic patterns and examine how new antimicrobials can be used appropriately as part of a broader AMR strategy. It is equally important to encourage the proper use of antivirals and other treatment options, supported by strong ASPs and better public awareness.

To achieve the UN General Assembly’s goal of 70% Access-group antibiotic use by 2030, Bangladesh will require coordinated efforts to reduce the overuse of Watch-category antibiotics. Also, investment in surveillance, education, and stewardship interventions across both the public and private healthcare sectors is essential. A multisectoral approach engaging policymakers, healthcare providers, pharmacists, and communities will be essential to curb AMR and safeguard the efficacy of existing antimicrobials for future generations.

## Figures and Tables

**Figure 1 antibiotics-14-00868-f001:**
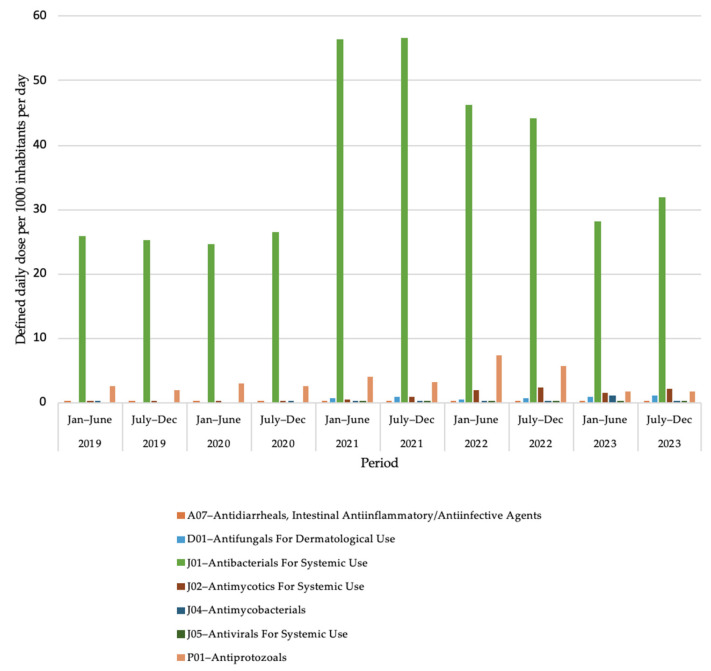
Antimicrobial use in Bangladesh from 2019 to 2023 at second level of Anatomical Therapeutic Chemical classification, based on defined daily dose per 1000 inhabitants per day.

**Figure 2 antibiotics-14-00868-f002:**
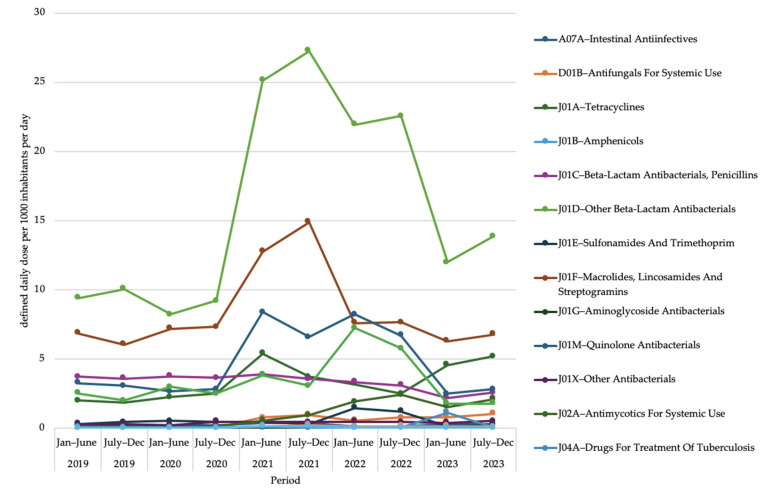
Antimicrobial use in Bangladesh from 2019 to 2023 at third level of Anatomical Therapeutic Chemical classification, based on defined daily dose per 1000 inhabitants per day.

**Figure 3 antibiotics-14-00868-f003:**
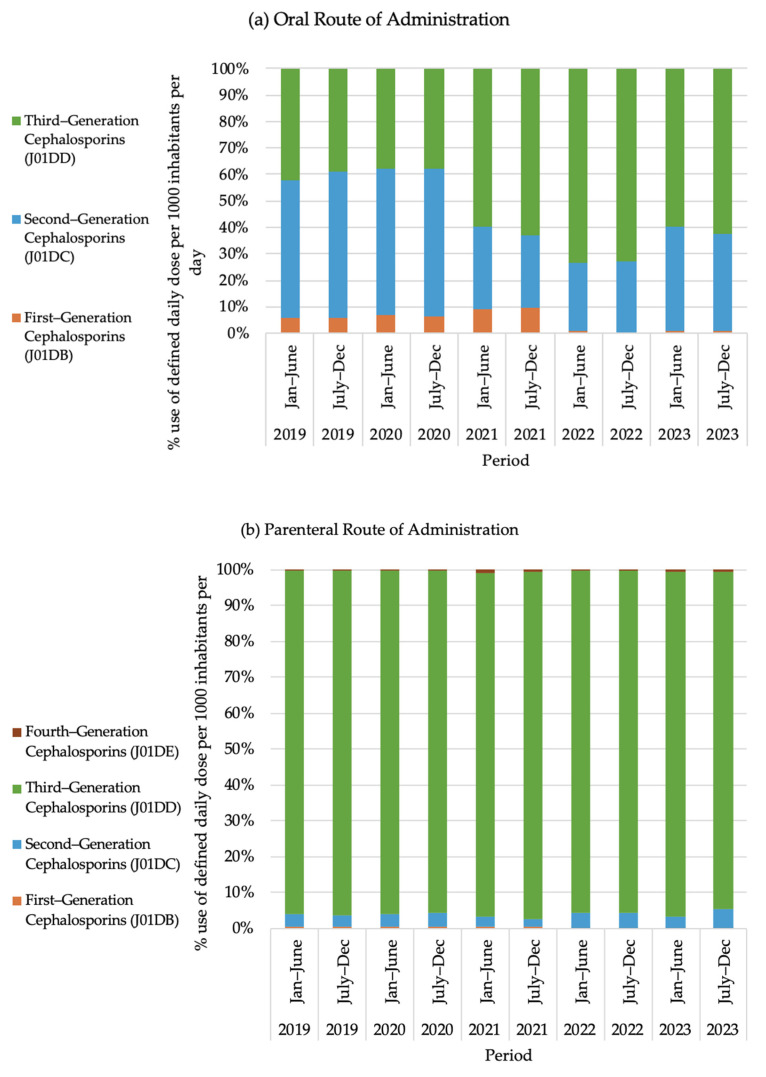
Cephalosporin use in Bangladesh from 2019 to 2023. (**a**) represents oral route of administration and (**b**) represents parenteral route of administration.

**Figure 4 antibiotics-14-00868-f004:**
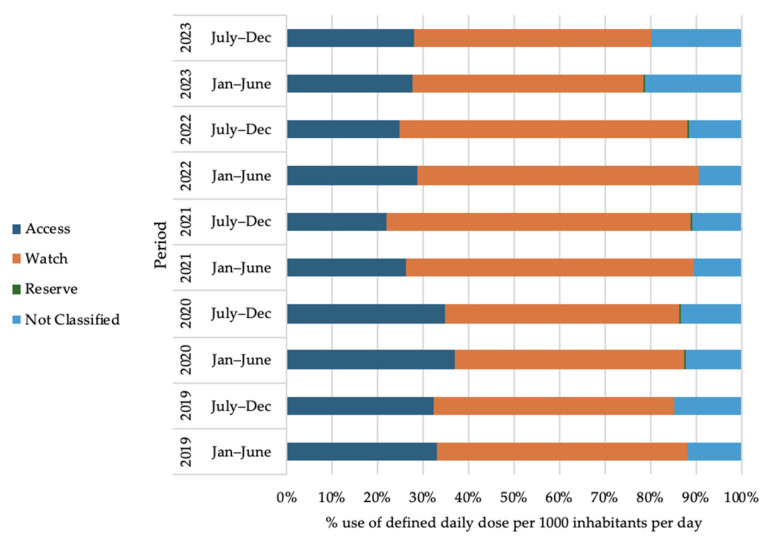
Oral antibiotic use by WHO AWaRe classification in Bangladesh (2019 to 2023).

**Figure 5 antibiotics-14-00868-f005:**
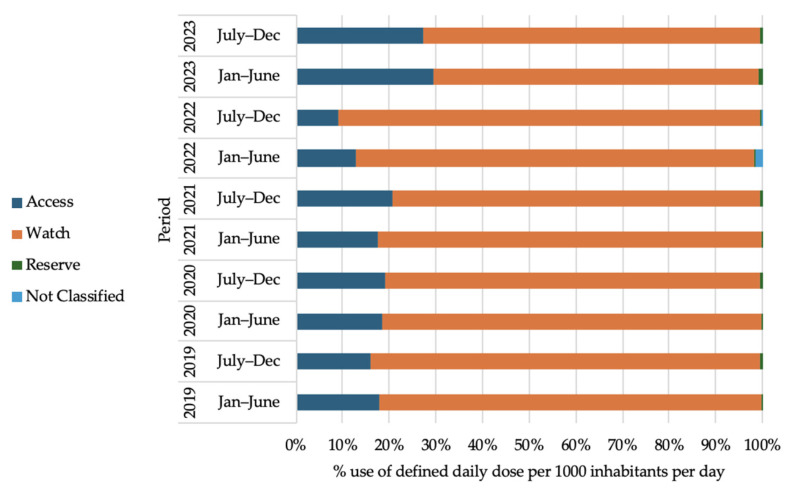
Parenteral antibiotic use by WHO AWaRe classification in Bangladesh (2019 to 2023).

**Table 1 antibiotics-14-00868-t001:** Overall pattern of national antimicrobial use in Bangladesh, according to defined daily dose per 1000 inhabitants per day (2019 to 2023).

Antimicrobial Medicine Use (Defined Daily Dose per 1000 Inhabitants per Day)										
	Pre–COVID-19 Period		During COVID–19 Period							After COVID–19 Period
Dosage Form	2019		2020		2021		2022		2023	
	Jan–June	July–Dec	Jan–June	July–Dec	Jan–June	July–Dec	Jan–June	July–Dec	Jan–June	July–Dec
Oral	28.02	26.97	27.45	28.74	60.91	61.30	54.67	52.13	33.01	36.49
Parenteral	0.55	0.58	0.46	0.46	1.38	1.06	1.48	1.16	0.73	0.88
Total	28.57	27.55	27.91	29.20	62.28	62.36	56.14	53.29	33.73	37.37

Sky blue color indicates the COVID-19 pandemic period.

**Table 2 antibiotics-14-00868-t002:** Most consumed oral antimicrobials covering 75% of national antimicrobial use (Drug Utilization 75%) in Bangladesh (2019 to 2023).

Percentage of Defined Daily Dose per 1000 Inhabitants per Day (Rank in Parentheses)										
Antimicrobial Agent (ATC Code)	2019		2020		2021		2022		2023	
	Jan–June	July–Dec	Jan–June	July–Dec	Jan–June	July–Dec	Jan–June	July–Dec	Jan–June	July–Dec
Cefixime (J01DD08)	12.7 (2)	13.17 (3)	10.24 (4)	11.18 (3)	21.48 (1)	24.57 (1)	27.66(1)	29.48 (1)	20.46(1)	22.26 (1)
Azithromycin (J01FA10)	23.31 (1)	21.22 (1)	25.24 (1)	24.57 (1)	19.91 (2)	23.18 (2)	12.72 (3)	13.24 (2)	18.01(2)	17.64 (2)
Cefuroxime and beta-lactamase inhibitor (J01DC52)	11.18 (3)	13.89 (2)	11.78 (2)	12.85 (2)	7.4 (5)	6.92 (4)	4.58 (6)	5.31 (5)	9.82 (4)	9.85 (4)
Ciprofloxacin (J01MA02)	6.66 (5)	6.62 (5)	5.65 (7)	5.62 (6)	10.56 (3)	7.83 (3)	11.98 (4)	10.3 (4)	4.09 (7)	4.42 (6)
Metronidazole (P01AB01)	8.77 (4)	7.06 (4)	10.59 (3)	8.45 (4)	6.22 (6)	4.88 (6)	13.12 (2)	10.9 (3)	5.26 (5)	4.72 (5)
Doxycycline (J01AA02)	6.39 (6)	6.05 (6)	7.28 (5)	7.78 (5)	8.68 (4)	5.92 (5)	5.45 (5)	4.41 (6)	13.38(3)	13.72 (3)
Cefuroxime (J01DC02)	5.14 (8)	5.29 (7)						3.73 (7)		
Flucloxacillin (J01CF05)		5.28 (8)		5.47 (7)					4.13 (6)	4.17 (7)
Amoxicillin (J01CA04)	5.41 (7)		5.97 (6)							
Cefradine (J01DB09)					3.66 (7)	4.09 (7)				
No of DU75 antimicrobials	N = 8	N = 8	N = 7	N = 7	N = 7	N = 7	N = 6	N = 7	N = 7	N = 7

Sky blue color indicates the COVID-19 pandemic period.

**Table 3 antibiotics-14-00868-t003:** Most consumed parenteral antimicrobials covering 75% of national antimicrobial use (Drug Utilization 75%) in Bangladesh (2019 to 2023).

Percentage of Defined Daily Dose per 1000 Inhabitants per Day (Rank in Parentheses)										
Antibacterial Agent (ATC Code)	2019		2020		2021		2022		2023	
	Jan–June	July–Dec	Jan–June	July–Dec	Jan–June	July–Dec	Jan–June	July–Dec	Jan–June	July–Dec
Ceftriaxone (J01DD04)	69.57 (1)	71.9 (1)	68.55 (1)	66 (1)	28.83 (2)	40.72 (1)	41.57 (1)	63.57 (1)	57.4 (1)	59.4 (1)
Moxifloxacin (J01MA14)					42.13 (1)	25.88 (2)	31.08 (2)	12.19 (2)		
Metronidazole (J01XD01)	7.84 (2)	6.88 (2)	8.92 (2)	8.16 (2)	9.31 (3)	10.44 (3)	7.68 (3)		5.47 (3)	4.97 (3)
Ciprofloxacin (J01MA02)					5.02 (4)					
Amikacin (J01GB06)			2.93 (3)	3.67 (3)						
Amoxicillin (J01CA04)									12.12 (2)	11.73 (2)
No of DU75 antibiotics	N = 2	N = 3	N = 3	N = 3	N = 4	N = 3	N = 3	N = 2	N = 3	N = 3

Sky blue color indicates the COVID-19 pandemic period.

**Table 4 antibiotics-14-00868-t004:** Number of antimicrobial medicines (generic) reported as consumed in Bangladesh (2019–2023).

No. of total antibiotics		96
1.1 Single-dose antibiotics	Total number	87
	Access	29
	Watch	44
	Reserve	10
	Not recommended	4
1.2 Fixed-dose combination antibiotics	Total number	9
	Access	3
	Watch	2
	Reserve	0
	Not recommended	4
2. No. of total antivirals	Total number	17
2.1 Single-dose antivirals		13
2.2 Fixed-dose antiviral combinations		4
3. No. of total antifungals	Single-dose	10
4. No. of total antiparasitics	Total number	9
4.1 Single-dose antiparasitics		7
4.2 Fixed-dose antiparasitics		2
5. No. of total antimicrobial medicines		132

## Data Availability

The data pertaining to this study is sharable and available from the corresponding author upon reasonable request.

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
