# Peer review of "Human Antimicrobial Use in Bangladesh: Five-Year Trend Analysis Including COVID-19 Pandemic Era"

_antibiotics, 2025, doi:10.3390/antibiotics14090868_

Round 1

Reviewer 1 Report

Comments and Suggestions for Authors

The article, "Human Antimicrobial Use in Bangladesh: Five-Year Trend Analysis Including Covid-19 Pandemic Era," addresses a critical public health issue, AMR, and provides valuable data from Bangladesh, where such national-level surveillance has been lacking. The methodology is sound, aligning with WHO guidelines. The temporal analysis, particularly the inclusion of the pre-COVID-19, during-COVID-19, and post-COVID-19 periods, offers important insights into the pandemic's influence on AMU patterns. The manuscript is well-structured and generally clear.

In Table 1 and other tables, the header rows for years and periods could be clearer for immediate understanding, though the content is accessible.

The labels on the x-axis of Figure 1 and Figure 2 are very dense. While readable, minor adjustments could improve readability without compromising information.

While the citation style is generally consistent, ensure every piece of information derived from a source is cited; for example, in the abstract, "cefixime and azithromycin being the most consumed" could also be cited if specifically drawn from the findings, although it's summarized there.

The discussion sometimes jumps between different aspects of AMU. Logical, ensuring smoother transitions between paragraphs might enhance readability. For instance, after discussing empiric antibiotic use in COVID-19, a clearer transition to the limited antiviral use could be beneficial.

The conclusion mentions the need for a longer post-COVID-19 period analysis for more comprehensive outline of change patterns. The use of new antimicrobials can be discussed and could be explicitly stated as a recommendation for future research. Other similar and recent work can help to build upon and enhance the conclusion part.

  • https://doi.org/10.3390/antibiotics10060738
  • https://doi.org/10.1371/journal.pone.0261368
  • https://doi.org/10.3390/antibiotics11050690

Author Response

Reviewer 1:

Journal: Antibiotics (ISSN 2079-6382)

Manuscript ID: antibiotics-3731326

Title: Human Antimicrobial Use in Bangladesh: Five-year Trend Analysis Including Covid-19 Pandemic Era

The article, "Human Antimicrobial Use in Bangladesh: Five-Year Trend Analysis Including Covid-19 Pandemic Era," addresses a critical public health issue, AMR, and provides valuable data from Bangladesh, where such national-level surveillance has been lacking. The methodology is sound, aligning with WHO guidelines. The temporal analysis, particularly the inclusion of the pre-COVID-19, during-COVID-19, and post-COVID-19 periods, offers important insights into the pandemic's influence on AMU patterns. The manuscript is well-structured and generally clear.

Comment 1: In Table 1 and other tables, the header rows for years and periods could be clearer for immediate understanding, though the content is accessible.

Response 1: We appreciate this helpful suggestion. We have revised the header rows of all relevant tables to make the periods clearer and improve immediate understanding for readers.

Comment 2: The labels on the x-axis of Figure 1 and Figure 2 are very dense. While readable, minor adjustments could improve readability without compromising information.

Response 2: Thank you for pointing this out. We have revised all the figures, including Figure 1 and Figure 2, to adjust the x-axis labels, making them less dense and more readable without losing any information.

Comment 3: While the citation style is generally consistent, ensure every piece of information derived from a source is cited; for example, in the abstract, "cefixime and azithromycin being the most consumed" could also be cited if specifically drawn from the findings, although it's summarized there.

Response 3: Thank you for your insightful comment. We acknowledge the importance of proper citation, even within the abstract when specific findings are referenced. The statement regarding "cefixime and azithromycin being the most consumed" is directly derived from the findings of our analysis, not from an external source. To avoid any ambiguity, we have revised the abstract to clarify that this information comes from our own results.

Comment 4: The discussion sometimes jumps between different aspects of AMU. Logical, ensuring smoother transitions between paragraphs might enhance readability. For instance, after discussing empiric antibiotic use in COVID-19, a clearer transition to the limited antiviral use could be beneficial.

Response 4: In response to your suggestion, we have revised the Discussion section to improve the logical flow and coherence between different aspects of AMU, especially during the COVID-19 pandemic. Specifically, we have added clearer transitions between paragraphs to avoid abrupt shifts. In particular, we included a smoother transition after the discussion on empiric antibiotic use to better introduce the topic of limited antiviral use. We also added linking phrases throughout to guide the reader and enhance overall readability.

Comment 5: The conclusion mentions the need for a longer post-COVID-19 period analysis for more comprehensive outline of change patterns. The use of new antimicrobials can be discussed and could be explicitly stated as a recommendation for future research. Other similar and recent work can help to build upon and enhance the conclusion part.

  • https://doi.org/10.3390/antibiotics10060738
  • https://doi.org/10.1371/journal.pone.0261368
  • https://doi.org/10.3390/antibiotics11050690

Response 5: Thank you for this constructive suggestion. We have carefully revised the conclusion to: (i) emphasize the importance of a longer post-COVID-19 period for future AMU analysis; (ii) explicitly recommend further investigation into the use of new antimicrobials in Bangladesh; and (iii) strengthen our conclusion, including the studies you recommended.

Reviewer 2 Report

Comments and Suggestions for Authors

Review report is attached.

Author Response

Reviewer 2:

 Journal: Antibiotics (ISSN 2079-6382)

Manuscript ID: antibiotics-3731326

Title: Human Antimicrobial Use in Bangladesh: Five-year Trend Analysis Including Covid-19 Pandemic Era

In this paper, authors presented the first national-level analysis of antimicrobial use (AMU) in Bangladesh across five years (2019-2023), including Covid-19 pandemic period. Authors utilized WHO’s defined daily dose (DDD), Anatomical Therapeutic Chemical (ATC) and Access, Watch, Reserve (AWaRe) classification methodologies. This topic has national and global relevance in the field of Antimicrobial resistance. However, the current draft of paper has some issues that needs to be addressed before it can meet publication standard.

Comment 1: Authors used central distribution data of antibiotics for the use. However, distribution does not directly correlate with patient consumption. Please explain what measures did authors utilized to measure and validate how much of distributed antimicrobials were actually consumed?

Response 1: We acknowledge the limitation that distribution data may not directly reflect actual patient-level consumption. In our study, we followed the methodology outlined by the WHO’s Global Antimicrobial Resistance and Use Surveillance System (GLASS), which recognizes aggregated data such as import, wholesaler, or central distribution data as valid proxies for estimating national antimicrobial consumption where patient-level data are not systematically available. According to GLASS Section 3.1, our data fall under “medicine-level AMU” (m-AMU) and were analyzed accordingly. While this does not capture individual patient use, it provides a reliable estimate for national trends. We have clarified this methodological approach in the revised manuscript to highlight both strengths and limitations of using aggregated data within the context of global surveillance.

Comment 2: Also, how did authors confirmed whether those distributed antimicrobials were expired, not taken or returned?

Response 2: Thank you for raising this important point. We addressed this through a multi-step data triangulation process: (i) we cross-checked manufacturers’ distribution data with IQVIA retail sales data to identify inconsistencies; (ii) manufacturers were instructed to exclude recalled, destroyed, or exported antimicrobials; and (iii) data were collected after each calendar year to incorporate final reconciliation with destruction and recall reports. These steps have now been described explicitly in the revised Methods section to clarify how we handled this potential bias.

Comment 3: Since authors used central distribution data, it would be interesting to see further categorical distribution of those antimicrobials based on demographics, region, location, private vs public hospitals or in-patient vs out-patients.

Response 3: We appreciate this valuable suggestion. Unfortunately, the central distribution data we used are aggregated at the national level and do not include stratification by demographics, region, sector, or patient setting. We agree that such disaggregation would provide important context and help tailor stewardship interventions. This has been noted as a priority recommendation for future surveillance systems and is now included in the revised Discussion section.

Comment 4: It is very interesting to see that Covid-19 pandemic increased the antimicrobial use. Is this increment due to immune conditions, any other underlying medical conditions or nation-wide policy. Please explain this more in the discussion.

Response 4: Thank you for this insightful comment. We have expanded the Discussion to explain that the increased AMU during the COVID-19 pandemic was likely driven by factors including empiric prescribing due to fear of co-infections, limited diagnostic capacity, high prevalence of underlying chronic conditions. Relevant references and clarifying context have been added to strengthen this explanation.

Comment 5: Authors analyzed data only until Dec 2023. Please include data until Dec 2024 to have more comparison post-covid time.

Response 5: We fully agree that including 2024 data would strengthen post-COVID-19 trend analysis. However, due to national reporting timelines, 2024 data are not yet finalized and would require significant additional time to collect, validate, and analyze. Including this would delay publication and go beyond our current study’s scope. We have added a note in the Conclusion highlighting the importance of including a longer post-COVID period in future research, as suggested.

Comment 6: Fix the numbering for Tables as Table 4 appears twice.

Response 6: Thank you for pointing this out. We have carefully revised the entire manuscript to ensure the correct numbering of all tables.

Comment 7: Remake Table 1-5 in tabular form with specific rows and columns to differentiate data properly and for better visualization.

Response 7: We have revised Tables 1–5 to improve clarity by adjusting headers, rows, and columns to clearly differentiate the data and enhance visual presentation.

Comment 8: Use better color separation for Figure 3-5.

Response 8: Thank you for this suggestion. We have improved the color schemes in Figures 3–5 to ensure clearer visual separation of categories and easier interpretation.

Comment 9: Use uniform formatting style for references.

Response 9: We have carefully reviewed and revised all references to ensure a consistent and uniform formatting style throughout the manuscript.

Comment 10: Put Figure 3a and 3b together. They are divided into two pages here.

Response 10: We have now combined Figure 3a and Figure 3b and ensured that they appear together on the same page in the revised manuscript.

Round 2

Reviewer 2 Report

Comments and Suggestions for Authors

The comments are satisfactory.